# Data-Driven Contact-Based Thermosensation for Enhanced Tactile Recognition

**DOI:** 10.3390/s24020369

**Published:** 2024-01-08

**Authors:** Tiancheng Ma, Min Zhang

**Affiliations:** Tsinghua Shenzhen International Graduate School, Tsinghua University, Shenzhen 518055, China; mtc21@mails.tsinghua.edu.cn

**Keywords:** data-driven algorithm, heat transfer modeling, quantitative thermosensation

## Abstract

Thermal feedback plays an important role in tactile perception, greatly influencing fields such as autonomous robot systems and virtual reality. The further development of intelligent systems demands enhanced thermosensation, such as the measurement of thermal properties of objects to aid in more accurate system perception. However, this continues to present certain challenges in contact-based scenarios. For this reason, this study innovates by using the concept of semi-infinite equivalence to design a thermosensation system. A discrete transient heat transfer model was established. Subsequently, a data-driven method was introduced, integrating the developed model with a back propagation (BP) neural network containing dual hidden layers, to facilitate accurate calculation for contact materials. The network was trained using the thermophysical data of 67 types of materials generated by the heat transfer model. An experimental setup, employing flexible thin-film devices, was constructed to measure three solid materials under various heating conditions. Results indicated that measurement errors stayed within 10% for thermal conductivity and 20% for thermal diffusion. This approach not only enables quick, quantitative calculation and identification of contact materials but also simplifies the measurement process by eliminating the need for initial temperature adjustments, and minimizing errors due to model complexity.

## 1. Introduction

High-precision tactile sensing is key for robotic systems to perform complex operations. Tactile signals comprise various signals, including force perception [1], sliding detection [2], thermosensation [3,4], comfort evaluation [5], and humidity detection [6]. Among these, thermal feedback is a crucial component and plays a significant role in autonomous control systems [7], virtual reality [8], and other fields [9].

Thermal conductivity and diffusivity coefficients are key parameters in thermal sensation. Various methods for measuring the thermal properties of solid and thin film materials have been developed and widely applied [10]. For solid materials, common techniques include the absolute method [11], comparative method [12], and parallel heat conduction method [13]. These steady-state methods accurately calculate thermal properties by precisely measuring the heat flux applied to the sample and the corresponding temperature difference. Compared to steady-state methods, transient methods can obtain thermal properties more quickly, typically relying on real-time temperature data and heat transfer models, such as the hot-wire method [14], laser flash method [15], and transient plane source method [16]. These methods offer high precision and stability but are often limited by the measurement mode, and have specific requirements for instruments and samples, rendering them unsuitable for real-time contact measurement applications in robotic systems.

In robotics and control systems, thermosensation increasingly focuses on miniaturization and flexibility. Traditional MEMS temperature sensors are known for precise design and packaging, effective in accurate temperature sensing; for example, the integrated temperature and pressure silicon chips for invasive endoscopic surgeries [17], the Schottky diode chip for temperature compensation [18], and thermistor chips in cyber-physical systems [19]. However, these silicon-based chips are typically unsuitable for irregular or dynamic surfaces and lack the conformability necessary for in-situ, human-like tactile sensing. Conversely, flexible devices can be attached to surfaces for in-situ thermosensation and researchers have extensively explored this area. For example, Zhao et al. [20] developed an intelligent finger integrated with a thermal sensor capable of measuring the thermal conductivity of objects upon contact, accurately distinguishing materials within 0.3 s. Li et al. [21] introduced a tactile sensor with a multi-layer structure; combined with machine learning algorithms, a robotic hand with ten sensors achieved a 94% recognition rate in waste sorting tasks. Yang et al. [22] proposed multi-layer tactile sensors integrated on a soft robotic hand, utilizing artificial neural networks to accurately identify 13 different materials under high contact pressures of 1.3–1.9 kPa. Lee et al. [23] developed an intelligent thermo-calorimeter (TCM) as a thermal sensing unit, successfully distinguishing various materials, especially metals, with high precision. Wu et al. [24] established a theoretical model based on the heat absorption process, allowing each material to be identified by a unique characteristic value.

The above methods achieve recognition through the extraction of temperature characteristics. Another approach involves using heat transfer models, and intelligent algorithms to directly calculate thermal properties. For example, Zhang et al. [25] employed an artificial neural network (ANN) to establish a correlation between thermal conductivity and its influencing factors, which is used to calculate the thermal conductivity of unsaturated soil. Fidan et al. [26] utilized neural networks with different neurons and activation functions to establish the relationship between the mechanical properties of concrete and its thermal properties. A coefficient of determination of 0.983 of thermal conductivity was achieved. Pan et al. [27] designed a thermosensation sensor and developed a heat transfer model for a robotic finger. Operating at room temperature (20 °C~32 °C), the system used an 8 mA current to generate Joule heat, accurately calculating the material’s thermal properties with a relative error within 10%. However, the heat transfer model is quite complex, and there is currently limited research on its quantitative thermosensation.

In this work, a novel contact-based thermosensation measurement method using different flexible thin-film thermal devices was designed to measure material’s heat flux and temperature. A data-driven algorithm, incorporating a discrete transient heat transfer model and a BP neural network, is proposed for processing measured signals and calculating thermal properties. The method’s accuracy and efficiency were experimentally validated, and it showed potential in material quantitative thermosensation and intelligent robot haptics.

## 2. Contact-Based Thermosensation Design

### 2.1. Design of the Measurement System Structure

Figure 1 presents a schematic diagram of the measurement system, which consisted of the test object, a sensing layer, a heating layer, and a substrate. The heater was designed to generate heat through Joule heating from an electrical current. A heat flux sensor, located in the center of the sensing layer and closely attached to the material’s surface, measured the heat flow across the material’s boundary. The sensor’s area was significantly smaller than the heating zone, and based on the semi-infinite assumption, it was considered that the heat would be transferred perpendicular to the object’s surface within a short time of the heater being activated. Temperature sensors were placed near the heat flux sensor to directly measure the surface temperature of the object.

The temperature of the tested material was influenced by both internal and external factors. Internal factors included the material’s thermal properties, while external factors mainly involved heat exchange between the system and the material. Consequently, modeling the sensing system was essential to establish how the temperature changes in different materials related to the heating power and the boundary heat flux.

### 2.2. Discrete Transient Heat Transfer Model

The heat flux signal, influenced by the heat source, cannot be expressed by a function, because the function fitting approach presents two significant challenges: first, the heat flux exhibits significant nonlinearity due to external excitation control, posing challenges to function fitting. Second, minor inaccuracies in the flux tend to accumulate progressively, resulting in cumulative errors over time.

Therefore, this study utilized a discrete approach to model heat transfer, aiming to minimize errors. The system was established to treat the object as semi-infinite, focusing on heat transfer perpendicular to the object’s surface and the transient process. Consequently, the model incorporated a non-stationary term and discretized both time and space simultaneously. In addition, this study explored the relationship between temperature, heat flux, source power, and thermal properties, simplifying the substrate to insulation boundary. For the heating layer’s upper boundary discretization unit, the discretization equation was as follows:(1)$$\left[-k_{h}\frac{T_{1}^{\left(i\right)}-T_{2}^{\left(i\right)}}{\Delta x}\right]\cdot \Delta \tau +q_{v}\cdot \frac{\Delta x}{2}\cdot \Delta \tau =\rho_{h}c_{h}\frac{\Delta x}{2}\cdot \left[T_{1}^{\left(i+1\right)}-T_{1}^{\left(i\right)}\right]$$
where $T_{n}^{i}$ denotes the temperature of the *n*-th discrete unit at time *I*; Δ*x* denotes the length of the discrete unit; Δ*τ* denotes the discrete time interval; *k_h_* is the thermal conductivity of the heater; *q_v_* is the power density of the heating unit; and *ρ_h_* and *c_h_* are the density and heat capacity of the heater, respectively.

The initial state of the system was steady state with temperature *T*_0_, denoted as follows:(2)$$\begin{array}{cc} T_{n}^{\left(1\right)}=T_{0} & n=1,2,\cdots ,N \end{array}.$$

Here, *N* denotes the total number of discrete units. Similarly, the sensing layer’s discrete control equation was formulated as follows:(3)$$k_{s}\frac{T_{n+1}^{\left(i\right)}-T_{n}^{\left(i\right)}}{\Delta x}+k_{s}\frac{T_{n-1}^{\left(i\right)}-T_{n}^{\left(i\right)}}{\Delta x}\begin{array}{cc} =c_{s}\rho_{s}\Delta x\cdot \frac{T_{n}^{\left(i+1\right)}-T_{n}^{\left(i\right)}}{\Delta \tau } & n=2,3,\ldots ,N_{1} \end{array}$$
where *k_s_*, *ρ_s_* and *c_s_* represent the thermal conductivity, density, and heat capacity of the sensing layer, respectively. *N*_1_ indicates the number of discrete units at the boundary beneath the sensing layer. For the boundary unit between the measured material and the sensing layer, the discrete equation was formulated as follows:(4)$$\begin{array}{cc} \left[k_{s}\frac{T_{\left(n-1\right)}^{\left(i\right)}-T_{\left(n\right)}^{\left(i\right)}}{\Delta x/2}+k_{o}\frac{T_{\left(n+1\right)}^{\left(i\right)}-T_{\left(n\right)}^{\left(i\right)}}{\Delta x/2}\right]\cdot \Delta \tau = & \rho_{o}c_{o}\Delta x\times \left[T_{\left(n\right)}^{\left(i+1\right)}-T_{\left(n\right)}^{\left(i\right)}\right]\\ & n=N_{1}+1 \end{array}$$
where *k_o_*, *ρ_o_* and *c_o_* represent the thermal conductivity, density, and heat capacity of the measured material. Subsequently, in accordance with the principle of energy conservation, the internal unit equation for the measured object was expressed as follows:(5)$$\begin{array}{cc} k_{o}\frac{T_{n+1}^{\left(i\right)}-T_{n}^{\left(i\right)}}{\Delta x}+k_{o}\frac{T_{n-1}^{\left(i\right)}-T_{n}^{\left(i\right)}}{\Delta x}= & c_{o}\rho_{o}\Delta x\cdot \frac{T_{n}^{\left(i+1\right)}-T_{n}^{\left(i\right)}}{\Delta \tau }\\ & n=N_{1}+2,N_{1}+3,\ldots ,N-1 \end{array}$$

Arranging the above equations resulted in a series of explicit iterative equations for the system:(6)$$\{\begin{array}{c} \begin{array}{cc} T_{n}^{\left(i+1\right)}=\frac{2\Delta \tau }{\rho_{h}c_{h}\Delta x}\cdot \left[\frac{k_{h}}{\Delta x}\left(T_{2}^{\left(i\right)}-T_{1}^{\left(i\right)}\right)+\frac{q_{v}\Delta x}{2}\right]+T_{n}^{\left(i\right)} & n=1 \end{array}\\ \begin{array}{cc} T_{n}^{\left(i+1\right)}=\frac{\alpha_{s}\Delta \tau }{\Delta x^{2}}\left[T_{n+1}^{\left(i\right)}+T_{n-1}^{\left(i\right)}\right]+\left(1-2\frac{\alpha_{s}\Delta \tau }{\Delta x^{2}}\right)T_{n}^{\left(i\right)} & n=2,3,\ldots ,N_{1} \end{array}\\ \begin{array}{cc} T_{\left(n\right)}^{\left(i+1\right)}=\frac{1}{\rho_{o}c_{o}\Delta x}\left[k_{s}\frac{T_{\left(n-1\right)}^{\left(i\right)}-T_{\left(n\right)}^{\left(i\right)}}{\Delta x/2}+k_{o}\frac{T_{\left(n+1\right)}^{\left(i\right)}-T_{\left(n\right)}^{\left(i\right)}}{\Delta x/2}\right]\cdot \Delta \tau +T_{\left(n\right)}^{\left(i\right)} & n=N_{1}+1 \end{array}\\ \begin{array}{cc} T_{n}^{\left(i+1\right)}=\frac{\alpha_{o}\Delta \tau }{\Delta x^{2}}\left[T_{n+1}^{\left(i\right)}+T_{n-1}^{\left(i\right)}\right]+\left(1-2\frac{\alpha_{o}\Delta \tau }{\Delta x^{2}}\right)T_{n}^{\left(i\right)} & n=N_{1}+2,\ldots \end{array},N-1\\ \begin{array}{cc} T_{n}^{\left(i\right)}=T_{0} & n=N \end{array} \end{array}$$
where *α_s_* and *α_o_* denote the thermal diffusion of the sensing layer and the tested material, respectively. The equation system was presented in an explicit differential format, allowing the computation of the temperature at each node for the moment directly subsequent from the initial temperature, thus eliminating the need for solving coupled equations. Programming the model in MATLAB facilitated the calculation of physical quantities.

### 2.3. Finite Element Simulation

COMSOL 6.0 was employed for a comparative analysis to validate the model. Given the structural features of the sensing system, rotationally symmetric modeling was adopted. A model representing one-sixth of the structure is depicted in Figure 2.

The simulation’s settings aligned with those in the MATLAB program. These included the geometric dimensions, with the radius of the heating layer, sensing layer, and the object set at 40 mm. The heat flux sensing area’s radius was 5 mm, the heater layer’s thickness was 20 μm, the sensing layer’s thickness was 0.4 mm, and the test material’s thickness was 40 mm. Additionally, the system’s initial temperature was set at 25 °C, and the heating power density of the heater was set at 7.70 × 10^7^ W∙m^−3^. Furthermore, the mesh was partitioned into highly refined triangular elements.

The thermal properties, as outlined in Table 1, were applied, setting the discrete unit spacing to 40 μm. Over a 2 s (experimentally chosen) duration, the solid’s heat transfer was calculated. Temperature and heat flux data from the sensing layer’s center, where it contacts the object, were used for comparison. Figure 2 displays the temperature distribution at time = 1.95 s, using “material III”. Figure 3 presents the results. The results indicated that the error between the model’s numerical calculations and the simulation is within 0.05 °C for temperature and within 30 W/m^2^ for heat flux, affirming the model’s effectiveness.

## 3. Data-Driven Algorithm

### 3.1. BP Neural Network

In machine learning, back propagation neural networks (BP NN) excel at learning and simulating relationships from the simple to the highly complex, rendering them ideal for various complex modeling and prediction tasks [28]. BP neural networks are not only fault-tolerant but also easier to implement compared to other machine learning methods, leading to their widespread application [29].

Serving as the core learning mechanism in multilayer feed-forward neural networks, the BP algorithm focuses on optimizing network weights via supervised learning. Essentially, the BP neural network is a multilayer nonlinear mapping structure, comprising neurons with varied connectivity weights. These neurons transform input data using activation functions, as depicted in Figure 4.

The error *E*, between the neural network’s actual output *y_i_*, and the expected output *Y*, was expressed as follows:(7)$$E=\frac{1}{2}\sum_{i}{\left(y_{i}-Y_{i}\right)}^{2}$$

Error propagated backward through the network, employing the gradient descent method to minimize the loss function and optimize network weights [30]. The fundamental weight update formula was:(8)$$w_{ij}^{n}=\Delta w_{ij}+w_{ij}$$

Here, *w_ij_* and *w_ij_^n^* represents the neuron’s current and updated weights, respectively, while Δ*w_ij_* denotes the change value, with its updated formula being as follows:(9)$$\Delta w_{ij}=-\eta \cdot \frac{\partial E}{\partial w_{ij}}$$

Here, *η* represents the learning rate, which determines the rate of weight adjustment. The essence of the BP algorithm is its utilization of the chain rule and gradient descent strategy to incrementally adjust the network, thereby aligning the network’s output more closely with actual observations. This iterative process continues until the output error becomes sufficiently small. Meanwhile, the loss function serves as a crucial tool for assessing the learning effect, quantifying the deviation between predicted and actual values, and acting as a key reference in the optimization process.

### 3.2. Data Set and NN Training

To accurately calculate thermal conductivities during testing, a mapping relationship between the material’s thermal conductivity, temperature, and heat flux was first established using a BP neural network. Using the proposed theoretical heat transfer model, the system’s heating power density (*q_v_*) was inputted to calculate the material’s boundary temperature (*T_f_*) and heat flux (*q_f_*) over a given time period, expressed as follows:(10)$$\left(T_{f},q_{f})=f_{h}(k_{o},\alpha_{o},t,q_{v}\right)$$
where *t* represents time, *k_o_* and *α_o_* are the thermal conductivity and thermal diffusivity of the tested material, and *f_h_* denotes the theoretical heat transfer model. In this study, 67 different types of metals and non-metals were selected [22,31,32], with thermal conductivity ranging from 0.06 to 405.5 W∙m^−1^∙K^−1^, covering the parameter range of common materials, as shown in Figure 5. The thermal properties of these 67 standard materials were inputted into the theoretical model program *f_h_* for calculation. The study focused on transient heat transfer over a period of 1.95 s, with a set power density range of 4.0 × 10^7^ W∙m^−3^ to 1.4 × 10^8^ W∙m^−3^, resulting in 1407 sets of time series data.

To enhance the training efficiency of the BP neural network, feature selection and data preprocessing were conducted on the training dataset. The specific steps were as follows:Feature extraction. To fully describe the characteristics of heat flux and temperature signals, the linear fitting slope of the heat flux relative to its initial value and the linear fitting slope of the temperature series were calculated, denoted as *u*_1_ and *u*_2_, respectively. The average heat flux and average temperature were calculated as *u*_3_ and *u*_4_. The final time’s excess temperature; the midpoint’s excess temperature; the temperature difference between the midpoint and final time; and the difference in heat flux are also calculated, respectively noted as *u*_5_, *u*_6_, *u*_7_, and *u*_8_.Normalization. The dataset covered materials ranging from low to high thermal conductivity, with corresponding heat flux and temperature data showing significant variations. Therefore, the data features were first natural log-transformed, then normalized and denoted as *X* = norm(ln(*u*)), resulting in *X* = [*x*_1_, *x*_2_, *x*_3_, *x*_4_, *x*_5_, *x*_6_, *x*_7_, *x*_8_].Principal component analysis (PCA). PCA is a data analysis technique that can retain as much of the original features as possible while reducing data dimensions [33,34]. By processing data with PCA dimensionality reduction, the principal components obtained were denoted as *p*_1_ to *p*_8_. The contribution rates of *p*_1_ and *p*_2_ exceeded 95%, indicating that *p*_1_ and *p*_2_ can explain over 95% of the variance in the original data, thus effectively representing the original feature. The relationship between *p*_1_, *p*_2_, and the original features is as follows:
(11)$$\{\begin{array}{l} p_{1}=-0.1961x_{1}+0.4242x_{2}-0.1974x_{3}+0.4242x_{4}+0.4244x_{5}\\ \;+0.4240x_{6}+0.4247x_{7}-0.1499x_{8}\\ p_{2}=0.5393x_{1}+0.1403x_{2}+0.5384x_{3}+0.1408x_{4}+0.1395x_{5}\\ \;+0.1417x_{6}+0.1378x_{7}+0.5669x_{8} \end{array}$$With *p*_1_ and *p*_2_ as inputs and the thermal conductivity *k_o_* as output, a double hidden layer nonlinear mapping network was trained using a BP neural network. In this network, the number of neurons in the input layer was 2, the first hidden layer contained 100 neurons, the second hidden layer contained 20 neurons, and the output layer contained one neuron. The tansig function was used as the activation function for the hidden layers.

### 3.3. BP NN with Heat Transfer Model

Based on the measured heat flux, temperature, and trained BP NN, the thermal conductivity of materials could be predicted. Furthermore, the calculation of thermal diffusivity could be studied. Taking the material under test as the subject of analysis, with heat flux as the input, and discretizing the object into *N* elements, the theoretical heat transfer model could be transformed into:(12)$$\left\{ \begin{array}{l} T_{1}^{\left(i+1\right)}=\frac{2\Delta \tau \alpha_{o}}{k_{o}\Delta x}\cdot \left[\frac{k_{o}}{\Delta x}\left(T_{2}^{\left(i\right)}-T_{1}^{\left(i\right)}\right)+q_{f}\right]+T_{1}^{\left(i\right)}\\ T_{N}^{\left(i\right)}=T_{0}\\ \begin{array}{cc} T_{n}^{\left(i+1\right)}=\frac{\alpha_{o}\Delta \tau }{\Delta x^{2}}\left[T_{n+1}^{\left(i\right)}+T_{n-1}^{\left(i\right)}\right]+\left(1-2\frac{\alpha_{o}\Delta \tau }{\Delta x^{2}}\right)T_{n}^{\left(i\right)} & n=2,\ldots \end{array},N-1 \end{array}\right.$$

Therefore, when the heat flux on the surface of the test object was used as input, the model could calculate the transient temperature on the material’s surface. This can be succinctly expressed as follows:(13)$$T_{f}=f_{q}(k_{o},\alpha_{o},q_{f})$$
where *f_q_* represents the transformed heat transfer model with heat flux as input, *q_f_* denotes the heat flux, and *T_f_* is the surface temperature of the object. Here, *q_f_* and *T_f_* are measured by sensors, *k_o_* is predicted by the BP NN model, and the function contains only one unknown variable. Thus, by scanning *α_o_* within a certain range, calculating the temperature and solving for the error with the actual measured temperature, the *α_o_* corresponding to the minimum error is searched as the result for the thermal diffusivity.

In summary, the proposed data-driven method for measuring thermal properties is illustrated in Figure 6. The algorithm combined forward calculation based on the heat transfer model to build a database, a BP neural network, and a transformed inversion solution for thermal diffusivity. By using the measured temperature and heat flux signals, it achieved quantitative thermosensation for different objects.

## 4. Experiment

### 4.1. Samples of Tested Materials

To validate the effectiveness of the proposed measurement method, three types of materials with smooth surfaces were chosen as experimental samples, as illustrated in Figure 7. These samples include tempered glass, polymethyl methacrylate (PMMA), and aluminum alloy, each measuring 80 mm × 80 mm × 20 mm. These materials, characterized by varying levels of thermal conductivity, are all homogeneous. Furthermore, it was necessary to accurately obtain the thermal properties of the sample materials as reference. To ensure accuracy, high-precision measuring instruments were used to test the samples, obtaining their density, specific heat capacity, and thermal diffusivity.

The specific experiments were as follows: (a) three types of solid samples were prepared, each cut to dimensions required by the analytical instruments. (b) The thermal diffusivities were quantified using the NETZSCH LFA447 Nanoflash analyzer. (c) The specific heat capacities were ascertained utilizing the Mettler-Toledo DSC3 differential scanning calorimeter. (d) The densities of the samples were determined through the displacement method, employing a precision balance. Finally, the results were recorded as shown in Table 2.

### 4.2. Experimental Measurement System

Based on the measurement method, a thermal sensation system was constructed, as shown in Figure 8. This system was utilized to verify the contracted based algorithm. The system could be specifically divided into three layers: the first layer consisted of a flexible heat flux sensor, temperature sensor, and flexible Polyimide (PI) film. The heat flux sensor used was the FHF05 series produced by Hukseflux company (Delft, Netherlands). The sensor was encapsulated in PI and outputted a voltage proportional to the heat flux. It was positioned between the test object and the serpentine heater, with a thickness of 0.4 mm and an area of 10 mm × 10 mm. Its sensitivity was 0.56 × 10^−6^ V/(W∙m^−2^), and had a measurement range of (−10~10) × 10^3^ W∙m^−2^.

The PI film served as a thermal resistance structure with the aim of aligning the thermal resistance perpendicular to the object’s surface. This alignment ensured that the temperature sensor’s heat transfer in the perpendicular direction shared identical thermal characteristics with the measurement area of the heat flux sensor. To attain this effect, a PI film matching the heat flux sensor in material and thickness, was chosen. The temperature sensor selected was a K-type fine-wire thermocouple (aidiwen, Yancheng, China). The probe part of the sensor had a thickness of 20 μm and was installed beneath the PI film. To fill the minute air gaps at the contact interface and between devices, thermally conductive grease (Arctic Silver 5) was used.

The system’s second layer, a serpentine heater, was positioned directly above the first layer. Comprising a serpentine metal electrode and a PI package, the heater generated Joule heat over a 55 mm × 60 mm area, with adjustable heating power achieved by varying the supply voltage.

Positioned above the heater, the third layer, or substrate layer, was made of 30 mm thick polyurethane (PU). During measurements, the heater supplied thermal energy, which was conducted to the object. Simultaneously, this thermal energy was transferred to the surrounding air. To minimize noise and disturbance, the substrate was made of a material with low thermal conductivity.

During the heating process, the heat flux sensor measured the heat flux signal across the boundary, while the thermocouple gauged boundary temperature in real-time, with both producing voltage output signals. A high-precision Keithley DAQ6510 digital multimeter was employed for voltage acquisition, configured with a 7700 series 20-channel plug-in switching module. This module, featuring built-in cold-end compensation circuitry, connected to K-type thermocouples and automatically converted the output to temperature readings. A Keithley 2280S-60 precision DC source supplied the input voltage to the heater. In addition, the sampling interval of measuring instruments differed from the algorithm’s time step, thus requiring adjustment through cubic spline interpolation.

## 5. Results and Discussion

Measurements were conducted using the developed thermal sensing system, yielding experimental data on temperature and heat flux, which was then processed using the proposed data-driven algorithm. To validate the effectiveness of the algorithm, the system’s power supply voltage was varied to adjust the heating power, resulting in different measured physical signals, which were subsequently analyzed.

Initially, the experiments focused on tempered glass, employing 12 V, 15 V, and 18 V heating voltages. The transient temperature and heat flux during heating were recorded, labeled as experimental data sets I, II, and III (refer to Figure 9).

Based on the experimental data of tempered glass, the algorithm was employed for signal processing, setting the *α_o_* scan range from 1 × 10^−7^ to 1 × 10^−6^ m^2^∙s^−1^ with a step size of 0.1 × 10^−7^ m^2^∙s^−1^. The algorithm’s calculated optimal curves and thermal properties are shown in Figure 10. The reference values were: *k_o_* = 1.097 W∙m^−1^∙K^−1^ and *α_o_* = 5.52 × 10^−7^ m^2^∙s^−1^ as shown in Table 2, with the calculated results having a relative error of less than 10%.

For PMMA, system voltages of 8 V, 15 V, and 20 V were used. The transient temperature and real-time heat flux during heating were recorded as experimental data sets I, II, and III, as depicted in Figure 11. Utilizing the experimental data, the algorithm processed the data with an *α_o_* scanning range of 1 × 10^−8^ to 5 × 10^−7^ m^2^∙s^−1^, and a step size of 0.05 × 10^−7^ m^2^∙s^−1^. The calculations yielded thermal conductivity and diffusivity, with algorithm-fitting curves presented in Figure 12. These values correspond to PMMA’s reference thermophysical properties: *k_o_* = 0.185 W∙m^−1^∙K^−1^ and *α_o_* = 1.10 × 10^−7^ m^2^∙s^−1^.

Figure 13 presents the surface heat flux and temperature measurements for the Al alloy, with supply voltages set at 15 V, 20 V, and 25 V, respectively. These results are respectively denoted as experimental data sets I, II, and III. The algorithmic computations and corresponding fitting curves are depicted in Figure 14, where the *α_o_* scanning range was set between 1 × 10^−5^ and 1 × 10^−4^ m^2^∙s^−1^, with a step increment of 0.1 × 10^−5^ m^2^∙s^−1^. Reference values for the Al alloy are as follows: *k_o_* = 120.4 W∙m^−1^∙K^−1^ and *α_o_* = 5.2 × 10^−5^ m^2^∙s^−1^.

Through experimentation, the method accurately calculates material thermal conductivity (*k_o_*) and diffusion coefficient (*α_o_*) across various materials, typically with less than 15% error under different heat power conditions. The algorithm simplifies the process by measuring only the surface excess temperature, eliminating the need for absolute temperature measurement and temperature compensation. This approach innovates in contact-based quantitative thermosensation.

## 6. Conclusions

This article introduces a contact-based thermosensation measurement method, focusing on the system design and signal processing algorithm. The system comprises a flexible heat flux sensor, a thermocouple, and a heater, facilitating the measurement of heat flux and real-time temperature at the material interface. A discrete transient heat transfer model was developed to establish the relationship between the heating power, heat flux, and temperature across various materials. The model was further analyzed and validated through finite element simulation. Building on this, a data-driven algorithm based on the heat transfer model and BP neural network was proposed. This algorithm calculates the thermal conductivity and thermal diffusivity of contacted materials through temperature and heat flux signals. In experiment, three different materials—PMMA, tempered glass, and aluminum alloy were measured under varying heating powers, yielding relative errors within 10% and 20% for thermal conductivity and thermal diffusivity, respectively. This method achieves enhanced thermosensation and holds potential for applications in intelligent robotic tactile sensing and automated control systems.

## Figures and Tables

**Figure 1 sensors-24-00369-f001:**
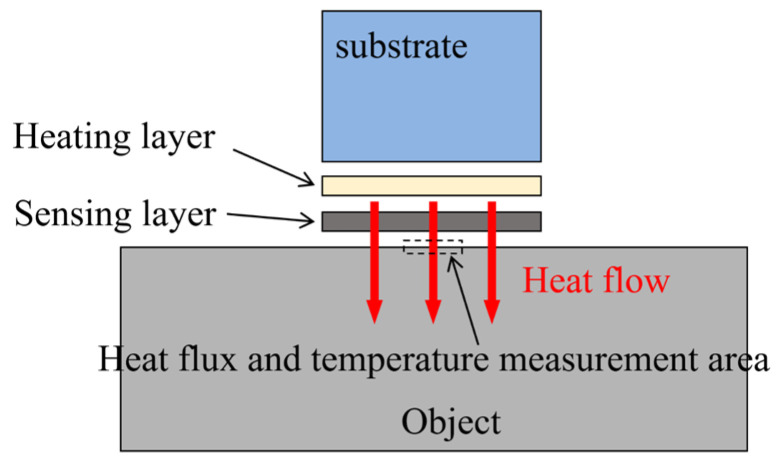
Schematic diagram for measuring heat flux and temperature in the system.

**Figure 2 sensors-24-00369-f002:**
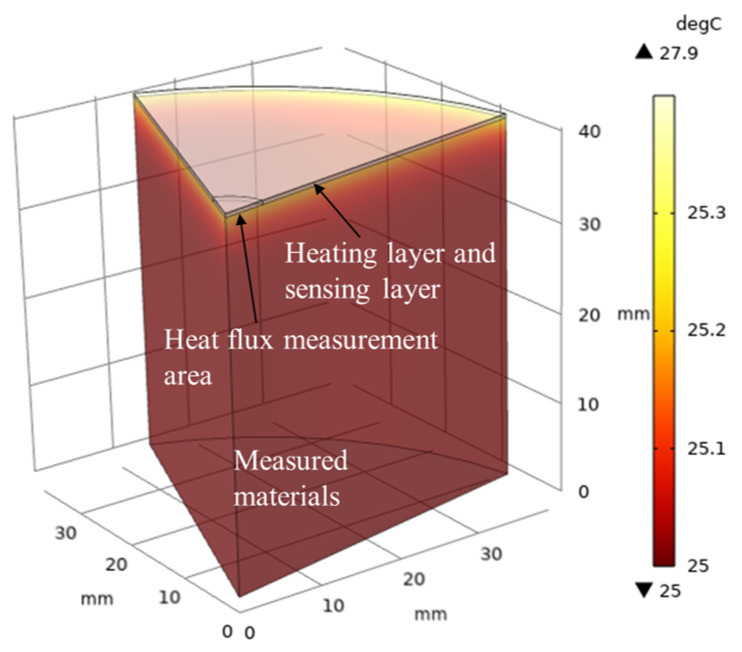
Heat transfer simulation model in Comsol.

**Figure 3 sensors-24-00369-f003:**
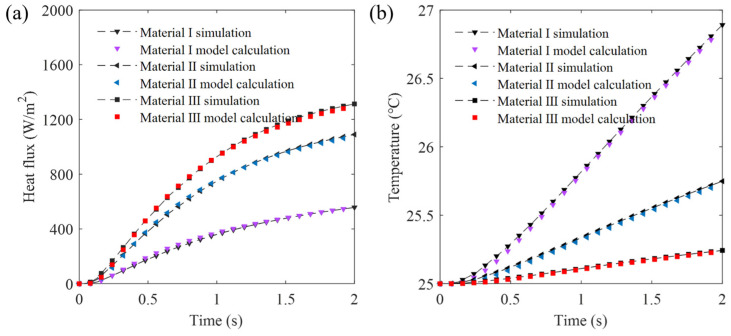
Comparison between the model and simulation (**a**) boundary heat flux and (**b**) temperature.

**Figure 4 sensors-24-00369-f004:**
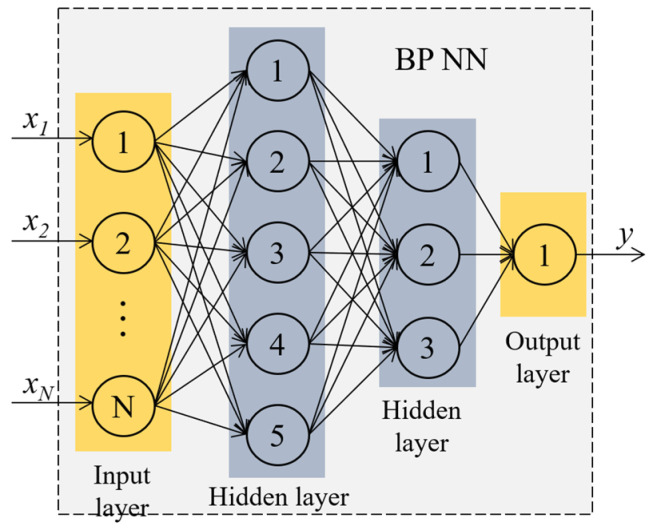
Conceptual schematic of the double hidden layer BP NN structure.

**Figure 5 sensors-24-00369-f005:**
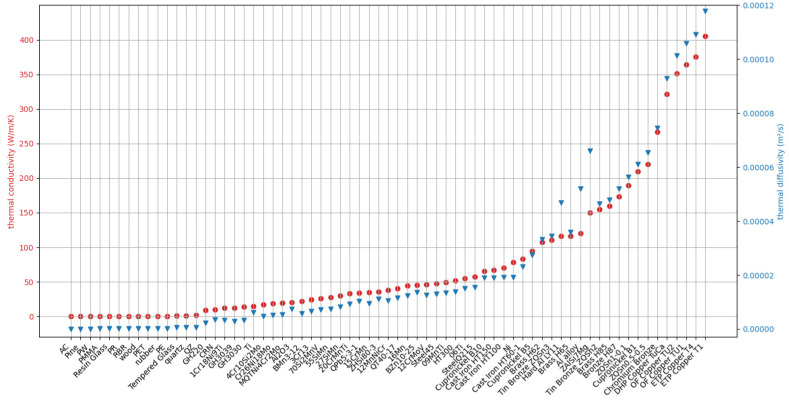
Thermal properties of different materials.

**Figure 6 sensors-24-00369-f006:**
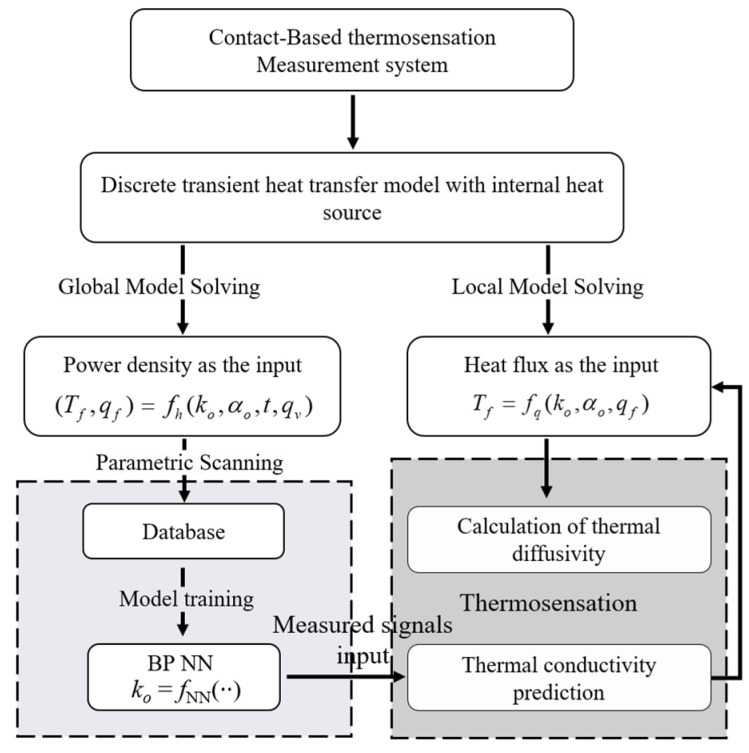
Flowchart of the data-driven algorithm for thermosensation.

**Figure 7 sensors-24-00369-f007:**
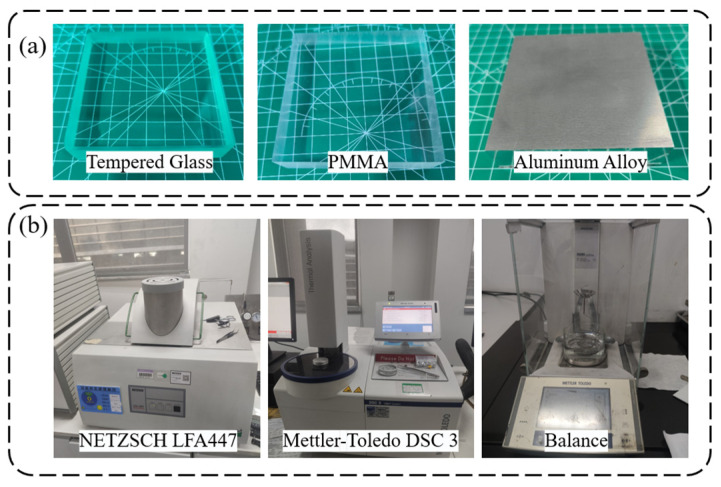
(**a**) Experimental samples and (**b**) instruments used for reference values measurement.

**Figure 8 sensors-24-00369-f008:**
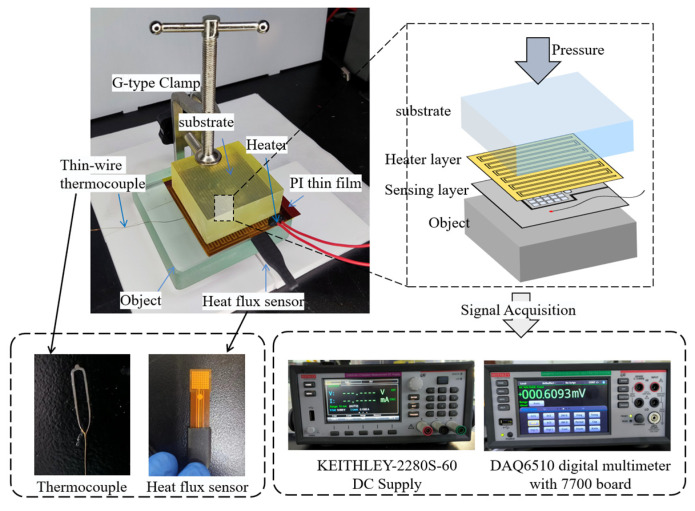
Experimental measurement system.

**Figure 9 sensors-24-00369-f009:**
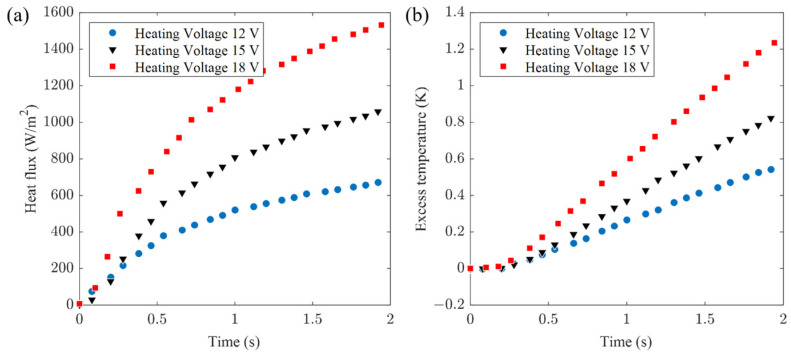
Experimental measurements on tempered glass (**a**) heat flux signal and (**b**) excess temperature.

**Figure 10 sensors-24-00369-f010:**
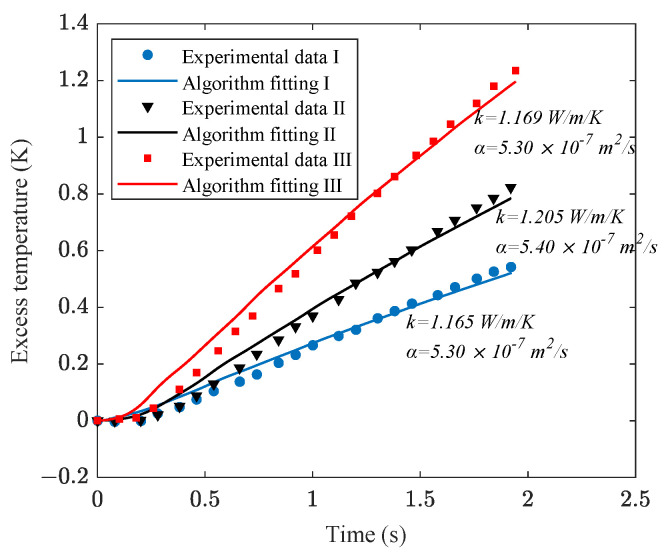
Experimental data and algorithmic calculation curve for tempered glass.

**Figure 11 sensors-24-00369-f011:**
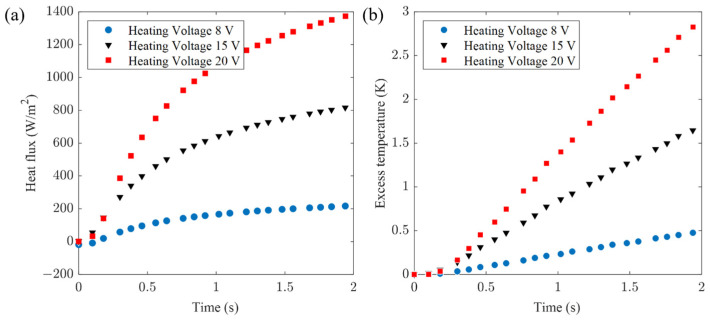
Experimental measurements on PMMA (**a**) heat flux signal and (**b**) excess temperature.

**Figure 12 sensors-24-00369-f012:**
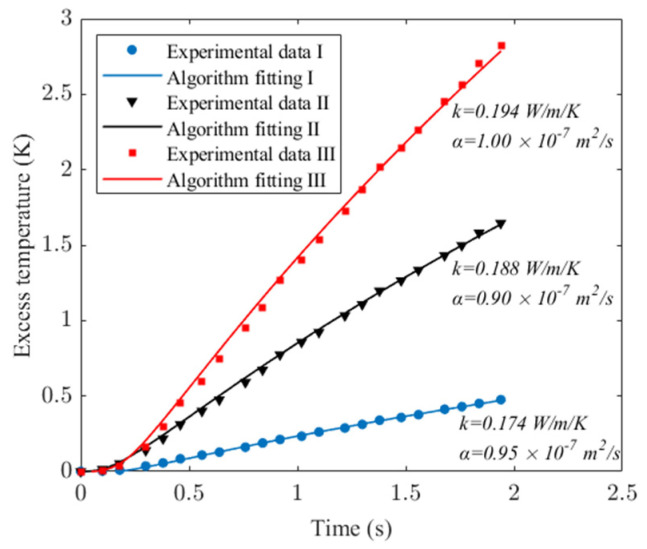
Experimental data and algorithmic calculation curve for PMMA.

**Figure 13 sensors-24-00369-f013:**
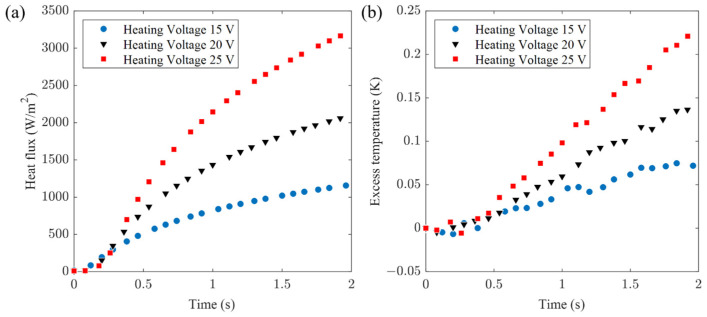
Experimental measurements on Al alloy (**a**) heat flux signal and (**b**) excess temperature.

**Figure 14 sensors-24-00369-f014:**
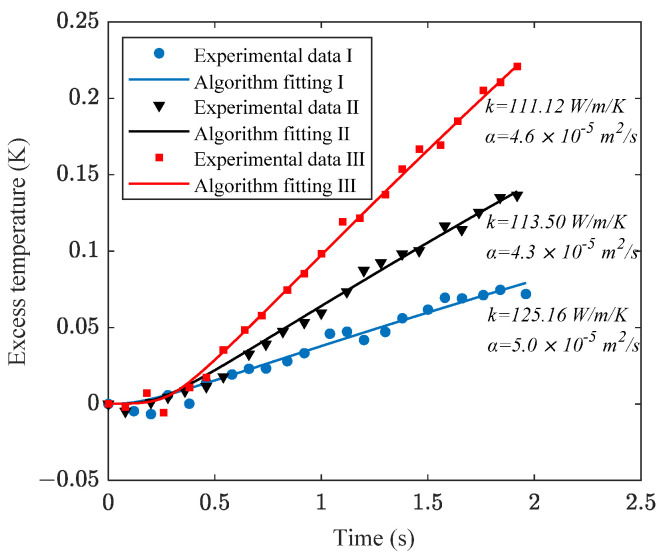
Experimental data and algorithmic calculation curve for Al alloy.

**Table 1 sensors-24-00369-t001:** Setting of thermophysical parameters in modeling and simulation.

Object	Thermal Conductivity/W∙m^−1^∙k^−1^	Density/kg/m^3^	Thermal Capacity /J/kg/K
Heating layer and sensing layer	0.214	1951.6	1064.6
Material I	0.1	500.2	2400
Material II	1.5	2659.6	800
Material III	12	7860	477.1

**Table 2 sensors-24-00369-t002:** Reference Measurements of Samples’ Thermal Properties.

Materials	Thermal Diffusivity /mm^2^∙s^−1^	Standard Deviation	Heat Capacity /J∙g^−1^∙K^−1^	Standard Deviation	Density /kg∙m^−3^	Standard Deviation
Tempered Glass	0.552	0.00350	0.809	0.0012	2458.4	5.798
PMMA	0.110	8.17 × 10^−4^	1.429	0.0029	1178.35	1.344
Aluminum Alloy	51.9	0.22	0.873	6.03 × 10^−4^	2651.5	17.82

## Data Availability

The data presented in this study are available on request from the authors.

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
