# Peer review of "Data-Driven Contact-Based Thermosensation for Enhanced Tactile Recognition"

_sensors, 2024, doi:10.3390/s24020369_

Round 1
Reviewer 1 Report
Comments and Suggestions for Authors
Dear authors,
Kindly go through my comments

Reviewer 2 Report
Comments and Suggestions for Authors
Dear Authors,
I appreciate the opportunity to review your important research in the field of temperature transfer or thermal sensitivity. This is an important development and its mechanism should certainly be applied globally. I still have a few questions and suggestions left, which will mainly add evidence to your manuscript. All comments are listed in the attached file "sensors-2798726-peer-review-v1 (Review)".
I wish you success in all your future research and I look forward to your soon reply.
Kind regards,
Reviewer

Round 2
Reviewer 1 Report
Comments and Suggestions for Authors
Done
Reviewer 2 Report
Comments and Suggestions for Authors
Dear Authors
Thank you for your response. I got all necessary answers. I wish you good luck in your future research.
Kind regards,
Reviewer